# Isolation and Evaluation of the Probiotic Activity of Lactic Acid Bacteria Isolated from Pickled *Brassica juncea* (L.) Czern. et Coss.

**DOI:** 10.3390/foods12203810

**Published:** 2023-10-17

**Authors:** Nguyen Hong Khoi Nguyen, Bach Long Giang, Tran Thanh Truc

**Affiliations:** 1Institute of Food and Biotechnology, Can Tho University, Can Tho 900000, Vietnam; nhknguyen@ntt.edu.vn; 2Faculty of Food and Environmental Engineering, Nguyen Tat Thanh University, Ho Chi Minh City 700000, Vietnam; 3Institute of Environmental Sciences, Nguyen Tat Thanh University, Ho Chi Minh City 700000, Vietnam; blgiang@ntt.edu.vn; 4School of Graduate, Can Tho University, Can Tho 900000, Vietnam

**Keywords:** *Brassica juncea* (L.) Czern. et Coss., isolation, lactic acid bacteria (LAB), probiotic activity

## Abstract

The naturally occurring lactic acid bacteria can be isolated from various sources. Pickled *Brassica juncea* (L.) Czern. et Coss. was used to isolate lactic acid bacteria (LAB). This study was conducted to compare the probiotic properties of probiotics isolated from pickled Vietnamese cabbage with some commercial strains of probiotics available on the Vietnamese market. The results showed that two strains (*Lactobacillus fermentum* and *Lactiplantibacillus plantarum*) isolated from pickled Vietnamese cabbage and three commercial strains of probiotics (*Bacillus subtilis*, *Bacillus clausii*, *Lactobacillus acidophilus*) all showed probiotic properties. Probiotic properties were evaluated through the ability to survive in low pH, pepsin, pancreatin, and bile salt media, the hydrophobicity of the bacteria, the antibiotic resistance, and the resistance to pathogenic bacteria. The isolated strain *Lactiplantibacillus plantarum* had fewer probiotic properties than *Bacillus subtilis* but more than the two commercial strains *Bacillus clausii* and *Lactobacillus acidophilus*, and the isolated *Lactobacillus fermentum* showed the fewest probiotic properties of the five strains.

## 1. Introduction

Lactic acid bacteria (LAB) are a member of the group of Gram-positive, non-spore-forming, non-respiratory, and rod-shaped cocci, which, through carbohydrate fermentation, produce lactic acid as the end product [1]. LAB are found in plants, insects, soil, water, and the human body [2]. In addition, LAB are also found in many fruits (grapes, olives, papaya, tomatoes, corn, etc.), vegetables, beans, and cereals [3]. LAB also participate in the main fermentation of more than 3500 fermented foods (kimchi, pickles, miso, yogurt, etc.), beverages, and silage (animal feed) [4]. *Brassica juncea* (L.) Czern. et Coss. is used in 3500 foods containing lactic bacteria because it is a traditional Vietnamese food that can be found everywhere. In addition, it is also a food that can be used directly without any processing. During fermentation, LAB produce bacteriocins that have antibacterial properties, fight pathogens, inhibit spoilage microorganisms, and contribute to the biological preservation of food [5]. At the same time, LAB use carbohydrates to create lactic acid and produce organic substances, which help to release odors, flavors, and aromas into the product. Furthermore, natural sources of LAB isolated in nature support the treatment of lactose intolerance, diabetes, obesity, acute diarrhea, enteritis, allergies, cardiovascular disease, and urinary tract infections and can prevent cancer [6]. The relationship between the gut microbiome and the brain is established in psychology as a two-way interaction system known as the Gut–Brain Axis [7]. To reinforce the effects of LAB on the brain, many clinical studies have been conducted, such as studies using *Lactobacillus* preparations to improve dementia, autism, and depression, showing that they have a positive impact on the performance of the human brain [8]. *Lactobacillus helveticus* R0052, *B. longum* R0175 [9], and *L. casei* [10] positively reduce stress and fatigue in humans and animals. In addition, some strains of LAB are considered probiotics and are defined as live microorganisms that exist in the intestinal tract with positive benefits for the host organism. The definition of probiotics does not refer to specific dosages but states that probiotics should be used in sufficient quantities to confer a health benefit to the host [11]. The study by Whorwell [12] reported that when examining probiotics at three different dosages (10^6^, 10^8^, and 10^10^ Colony Form Units per milliliter (CFU/mL)), the results showed that the dose of 10^8^ CFU/mL was the most effective compared to the other two doses.

Currently, lactic acid strains are widely used in daily life and commercialization. However, during processing in adverse conditions, spray drying, mechanical stress, dehydration, heating, and oxygen exposure can lead to decreased LAB cell activity and viability [13]. In general, probiotics have achieved recent development and growth in dietary supplements, but only a few studies evaluate their safety, efficacy, and comparative activity [14]. The purpose of this study is to isolate LAB from pickled *Brassica juncea* (L.) Czern. et Coss. and compare the probiotic activity of potential probiotic strains with existing commercial probiotic strains on the Vietnamese market by assessing the viability of the potential probiotics in low pH, pepsin, pancreatin, and bile salt media, the hydrophobicity of the bacteria, and their antibiotic sensitivity and resistance to harmful bacteria.

## 2. Materials and Methods

### 2.1. Materials and Chemicals

#### 2.1.1. Material

*Lactobacillus* bacteria in pickled *Brassica juncea* (L.) Czern. et Coss. and microbial products were produced at the International Medical and Pharmaceutical Consulting Co., Ltd., Sunbiozyms biotechnology factory (Lot 38-2, Quang Minh I Industrial Park, Quang Minh Town, Me Linh District, Hanoi City): *Bacillus subtilis*, *Bacillus clausii*, *Lactobacillus acidophilus*.

#### 2.1.2. Chemicals

*Lactobacillus* MRS agar (Himedia, India), Mueller Hinton Agar/Broth (Himedia, India), and microbial preparations were produced at the International Medical and Pharmaceutical Consulting Co., Ltd., Sunbiozyms biotechnology factory (Lot 38-2, Quang Minh I Industrial Park, Quang Minh town, Me Linh district, Hanoi City, Vietnam), including *Bacillus subtilivids* 1.5B containing *Bacillus subtilis* spores with a spore count ≥ 1.5 × 10^9^ CFU/g; bacterial preparations of *Bacillus Clausivids* 1.5B containing *Bacillus clausii* spores with spores ≥ 1.5 × 10^9^ CFU/g; bacterial preparations of *L. acidophilus* VIDS 3B containing *Lactobacillus acidophilus* spores ≥ 3 × 10^9^ CFU/g. NaCl 99.5% (China), CaCl_2_ 98% (China), Na_2_HPO_4_·12H_2_O 99% (China), KCl 99.5% (China), KH_2_PO_4_ ≥ 99.5% (China), NaHCO_3_ 99.5% (China), CH_3_COONa·3H_2_O 99% (China), HCl 36–38% (China), glycerol 99% (China), xylene 99% (China), lipase (ICFOOD Vietnam), pepsin B1 digestive enzyme (Vietnam), pancreatin (China), bile salt (China), trypsine (China), alcohol 96°, bromocresol green reagent (China), Gram stain kit (China).

### 2.2. Methods

#### 2.2.1. Isolation, Morphological Characterization, Biochemical Testing, and Gene Sequencing of Probiotic Strains

*Brassica juncea* (L.) Czern. et Coss. were collected at different markets in Ho Chi Minh City, Vietnam. After purchasing, *Brassica juncea* (L.) Czern. et Coss. (1 kg) was washed and then soaked in a 3% salt water solution (2 L) and 0.25% sugar was added. After 3 days, the cabbage began to turn yellowish and the fermented juice became cloudy. Briefly, 10 mL of pickled *Brassica juncea* (L.) Czern. et Coss solution was homogenized with 90 mL of 0.9% saline solution and serial diluted to concentrations of 10^−2^, 10^−3^, 10^−4^, and 10^−5^. Following this, 100 µL of diluent was spread on the surface of the MRS agar. The plates were incubated at 37 °C for 48 h. Subsequently, colonies of different shapes were streaked on MRS agar to purify the strains. Then, bacterial strains were observed morphologically by the Gram staining method and biochemical properties (catalase, acid production, glucose utilization) to screen for LAB. Finally, bacteria were identified by 16S rRNA gene sequencing. Sequences were analyzed using the database of GenBank and the BLAST (http://www.ncbi.nlm.nih.gov/BLAST, 15 August 2023).

#### 2.2.2. Determination of Low pH Tolerance in Probiotic Strains

When producing biological products, it is necessary to select strains that can withstand low pH media for at least 90 min and survive at pH 3 for 2 h with an acid concentration of 1000 mg/L. This is the optimal acid tolerance for probiotic strains [15]. Bacterial strains were proliferated on an agar medium and the bacterial biomass was collected and washed in 0.85% saline solution. The bacterial suspension was diluted in a PBS medium and the initial number of viable cells was assessed. The viability of probiotics in a low pH environment was determined by 2 mL of the suspension added to test tubes containing 2 mL of PBS solution at pH 1, pH 2, and pH 3. They were incubated at 37 °C (each pH value was repeated 3 times). After 0 min, 30 min, 90 min, and 180 min of incubation, 0.1 mL of bacterial culture was aspirated onto the MRS agar plate and the number of bacteria was checked by the colony counting method.

#### 2.2.3. Determination of the Survival Ability in the Simulated Pepsin Media of Probiotic Strains

Microbial strains were propagated on an agar medium and the biomass was washed with 0.85% NaCl solution and then centrifuged. The viability of probiotics in pepsin media was determined using simulated gastric juice in which pepsin (3 g/L) was dissolved in sterilized saline (NaCl 0.5% *w*/*v*) and then adjusted to pH 2.0 with 1 M HCl. Then, 9 mL of simulated gastric juice was placed in test tubes containing 1 mL of suspension. The survival rate of the probiotics was determined by the number of colonies grown on MRS agar after 0 min, 30 min, 90 min, and 180 min incubation.

#### 2.2.4. Determination of Viability in the Simulated Pancreatin Media of Probiotic Strains

The viability of probiotics in the pepsin media was determined using simulated pancreatic juice in which pancreatin (1 g/L) was dissolved in sterile saline (0.5% *w*/*v* NaCl) and then adjusted to pH 7.0 with 0.1 M NaOH at 37 °C [16]. Then, 9 mL of simulated pancreatic juice (addition of 0.45% bile salts) was added to test tubes containing 1 mL of the suspension. The survival rate of the probiotics was determined by the number of colonies growing on the MRS agar plate after 0 h, 1 h, 2 h, 3 h, and 4 h incubation.

#### 2.2.5. Determination of the Ability to Survive in the Simulated Bile Salt Media of Probiotic Strains

Bile salts are produced from cholesterol metabolism and are involved in digestion to break down and absorb fats. Furthermore, bile salts also prevent the existence of microorganisms in the intestine. In humans, the mean concentration of bile salts in the small intestine is 0.3% [17]. To evaluate the acid-bile tolerance of probiotic strains, the microbial strains, after being purified and proliferated at certain concentrations, were incubated in PBS buffer (pH 2) with added cholera (0.3% *v*/*v*). They were incubated for 1 h, 2 h, 3 h, and 4 h using a culture method to determine the remaining microbial density after incubation.

#### 2.2.6. Determination of the Hydrophobicity of Probiotic Strains

The hydrophobicity of the probiotics was determined based on the method of Vinderola [18]. Initially, LAB (3 commercial and 2 potential strains) were grown in an MRS liquid media at 37 °C for 24 h. The bacterial cell suspension was centrifuged at 15,000 rpm for 15 min, removing the cytosolic supernatant. After this, the bacteria were resuspended in PBS buffer (pH 7.0) to obtain an optical density value OD_600nm_ = 0.4–0.6 (*A*_0_). Next, 5 mL of xylene solvent was added to the test tubes, vortexed for 3 min, and allowed to settle for 1 h. Then, delamination occurred, the hydrophobicity was removed, the bacterial suspension solution was collected, and the optical density value OD_600nm_ was measured a second time (*A*_1_). This formula determines the hydrophobicity of bacteria:

%$H=\frac{A_{0}-(A_{1}-A_{Blank})}{A_{0}}\times 100$ in which *A*_0_ is an optical density value OD_600nm_ = 0.4–0.6; *A*_1_ is the optical density value OD_600nm_ measured after 1 h; and *A*_blank_ is the optical density value of the sample without the bacteria and hydrocarbon supernatant.

#### 2.2.7. Determination of the Antibiotic Resistance of Probiotic Strains

The antibiotic susceptibility test of probiotic strains was performed using the diffusion method in agar plates [19]. Briefly, probiotic strains were grown in MRS liquid (*Lactobacillus fermentum*, *Lactiplantibacillus plantarum*, *Lactobacillus acidophilus*) and TSB liquid (*Bacillus subtilis*, *Bacillus clausii*) at 37 °C for 24 h. Following this, 100 μL of probiotic strains (1.5 × 10^8^ CFU/mL) were spread on the surfaces of the respective agar plates. Subsequently, paper discs containing antibiotics (d = 6 mm) with defined concentrations were placed on the petri dish. The antibiotics discs consisted of *Amoxicillin* (Ax) 10 μg, *Doxycycline* (Dx) 30 μg, *Enrofloxacin* (Ef) 5 μg, and *Gentamicin* (Ge) 10 μg as per the standard of the CLSI (2015). Then, the plates were incubated at 37 °C for 24 h and the antibacterial circle diameters were recorded on the agar surface.

#### 2.2.8. Determination of the Resistance of Pathogenic Bacteria to Beneficial Bacteria

The antibacterial test of probiotic strains against 10 pathogenic bacteria using diffusion on agar plates was based on the method of Liu [20]. Initially, pathogenic and probiotic strains were cultured in MHB and MRS liquid at 37 °C for 24 h. Following this, 100 μL of pathogenic bacteria (1.5 × 10^8^ CFU/mL) were spread on the surface of the MHA agar. Then, 50 μL of the probiotic strains were added into 6 paper wells (d = 6 mm) on the surface of the agar plate. Ciprofloxacin was used as a positive control. The plates were incubated at 37 °C for 24 h. After the incubation, the diameters of the inhibitor zones on the agar dish were measured and recorded.

#### 2.2.9. Statistical analysis

The results of the comparison and selection experiments were statistically processed and analyzed using the IBM SPSS Statistics 20 program. Analysis of variance (ANOVA) and post hoc Tukey’s test were used for group comparisons. The level of statistical significance was set at *p* < 0.05. Data are presented as the mean ± standard deviation (SD).

## 3. Results

### 3.1. Isolation of Probiotic Bacteria from Pickled Brassica juncea (L.) Czern. et Coss.

The colony morphology, bacterial morphology, and biochemical characteristics of six strains of bacteria (N1, N2, N3, N4, N5, N6) isolated from *Brassica juncea* (L.) Czern. et Coss. are shown in Table 1. Table 2 showed that, in general, the colonies were ivory-white or yellow, but the size of the colonies varied between strains. Table 3 showed the detailed information of the Gram staining results showed that N1, N2, N3, N4, and N6 strains belong to Gram-positive bacteria, among which the N2 strain had a long rod shape, and the remaining strains had a short rod shape. The N5 strain belonged to Gram-negative bacteria. Moreover in Table 4, biochemical tests determined that the N2 and N6 strains could not produce catalase but could use metabolized glucose to produce acid. Table 5 presented the identification results by the 16S rRNA sequencing method, it showed that the gene from the N2 strain was 100% similar to the gene encoding the 16S rRNA of the *Lactobacillus fermentum* strain (accession number MT611892.1) and the N6 strain had 100% similarity with the *Lactiplantibacillus plantarum* strain (accession number OQ891485.1). Appendix A showed genome sequence of two potential probiotics.

### 3.2. Viability of the Potential Probiotics Compared with Commercial Strains in Low pH Media

The percentage of bacteria that survived at 0 min, 30 min, 90 min, and 180 min in the media at pH 1, 2, and 3 is shown in Table 6. At 180 min at pH 2 and pH 3, the probiotic strains that had the highest viability were *B. subtilis* and *B. clausii*, respectively, and *L. fermentum* was the lowest existing strain. However, at pH 1 and 180 min, all five strains of probiotics could not survive. This result was like the study of Thankappan [21] when studying the characteristics of *Bacillus* spp. At pH 1, all potential probiotics could not survive, and this was explained by the fact that at pH 1, the probiotics were damaged and died [22]. On the other hand, when a probiotic strain was compared for each incubation period of 30 min, 90 min, and 180 min and at three different pH values, it was shown that at pH 1 and an incubation time of 180 min, the survival ability of the probiotics was the lowest; at pH 3 and an incubation time of 30 min, the viability of all five strains of probiotics reached the highest value, respectively, *B. subtilis*, *L. acidophilus*, *L. plantarum*, *B. clausii*, and *L. fermentum*.

### 3.3. Survival Ability of the Potential Probiotics Compared with Commercial Strains in Pepsin-Containing Media

The percentage rate of vital bacteria at 30 min, 90 min, and 180 min in pepsin media is shown in Table 7. The results show that all five strains of beneficial bacteria were able to survive in pepsin media (3 g/L, pH 2 for 180 min) and are arranged in descending order as *L. acidophilus* 47.45% ± 0.22, *B. subtilis* 43.19% ± 0.05, *L. plantarum* 41.16% ± 0.14, *B. clausii* 31.22% ± 0.04, and *L. fermentum* 18.20% ± 0.06. The viability in the pepsin media of isolated *L. plantarum* was lower than that of the *L. acidophilus* commercial strain but higher than that of *B. subtilis*, *B. clausii*, and *L. fermentum*. Similarly, in the study of Kaewnopparat [23] and Kang [24] on *L. fermentum* SK5 and *L. salivarius* probiotic strains in pepsin (3 g/L, at pH 3, pH 4), the results showed that after 3 h and 4 h of incubation, the survival rate of probiotics reached more than 99% or was barely affected. At pH 2, for the *L. fermentum* SK5 strain, the survival rate decreased after 3 h of incubation, and the number of viable cells reached 70.48%, while for the *L. salivarius* strain after 4 h, the survival rate was 85%. Thus, when the incubation time in the media with pepsin at pH 2 was 3 h or 4 h, the percentage of viable cells compared to the original decreased.

### 3.4. Viability of the Potential Probiotics Compared with Commercial Strains in Pancreatin Media

The percentage of bacteria that survived at 1 h, 2 h, 3 h, and 4 h in the media containing pancreatin is shown in Table 8. The results of this experiment showed that all five strains of probiotics were able to survive in pancreatin medium after 4 h of incubation, but the reduction rate was not much and they were arranged in descending order as *B. subtilis* (97.13% ± 0.07), *L. plantarum* (96.2% ± 0.11), *L. fermentum* (94.36% ± 0.04), *L. acidophilus* (94.27% ± 0.07), and *B. clausii* (93.15% ± 0.04). This showed that the viability of isolated *L. plantarum* in a simulated pancreatin media was lower than the *B. subtilis* commercial strain but higher than two other commercial strains, *L. acidophilus* and *B. clausii*.

### 3.5. Ability of the Potential Probiotics Compared with Commercial Strains in Bile Salt Media

The results of this experiment showed that all five strains of probiotics could survive in the bile salt media, as shown in Table 9. Like the results when compared in pancreatin medium, the probiotic strains had a survival rate that decreased gradually with incubation time and after 4 h of incubation, the *B. subtilis* strain was able to survive the longest at 64.17% ± 0.05 and the *L. fermentum* strain had the lowest survival in bile salt media with 39.27% ± 0.05. In general, all five probiotic strains survived in 0.3% bile salt, but after 4 h of incubation, the number of cells tended to decrease.

### 3.6. Investigation of the Hydrophobicity of Potential Probiotics Compared with Commercial Strains of Probiotics

The hydrophobicity of probiotics was calculated based on the ability to reduce the absorbance of the cell suspension. The hydrophobic capacity results of the five bacterial strains are shown in Table 10. The research results show that three commercial (*L. acidophilus*, *B. subtilis*, *B. clausii*) and two isolated strains (*L. fermentum*, *L. plantarum*) were able to adhere to the intestinal epithelium with hydrophobic properties with hydrophobicity (xylene) at a relative level. The hydrophobicity of probiotic strains is arranged in decreasing order as *Bacillus subtilis* > *Lactiplantibacillus plantarum* > *Bacillus clausii* > *Lactobacillus acidophilus* > *Lactobacillus fermentum*. Based on the research results, the commercial *Bacillus subtilis* strain showed the highest hydrophobicity with an efficiency of 40.42%. The *L. plantarum* strain isolated from pickled mustard greens also showed good adhesion (efficiency of 30.95%), higher than that of commercial *Bacillus clausii* and *Lactobacillus acidophilus*.

### 3.7. Antibiotic Susceptibility of Isolated Compared to Commercial Strains of Probiotics

The antibiotic susceptibility of probiotic bacteria is often of interest in probiotics because it represents potential biosafety. Antibiotic-susceptible strains do not contain antibiotic resistance genes of the antibiotic resistance lines. This study shows the antibiotic susceptibility results of five probiotic strains (Table 11). Experimental images are included in Appendix B.

The study showed that all five bacterial strains were sensitive to three antibiotics (*Doxycycline*, *Enrofloxacin*, and *Amoxicillin*). The inhibitor zone diameter was in the range of 29.67 ± 0.577–30.33 ± 1.528 mm (*Doxycycline*), 23.00 ± 1.00–30.67 ± 1.528 mm (*Enrofloxacin*), and 22.67 ± 1.155–28.67 ± 0.577 mm (*Amoxicillin*), more considerable than the diameter at the sensitivity limit (S) of each antibiotic prescribed by CLSI M100-S21, specifically *Doxycycline* (S ≥ 14 mm), *Enrofloxacin* (based on the standard of antibiotics of the same class, S ≥ 21 mm), and *Amoxicillin* (S ≥ 17 mm). Among these, the diameter of the resistance zones produced by bacteria *B. subtilis*, *L. plantarum*, and *B. clausii* to *Enrofloxacin* antibiotics differed from that of the two strains *L. acidophilus*, and *L. fermentum*. For the antibiotic *Amoxicillin*, the diameter of the inhibitor zone of *B. subtilis* was significantly different from that of the other bacteria, and the diameters of the three strains of *L. acidophilus*, *B. clausii*, and *L. plantarum* were not significantly different when compared with each other, but they were significantly different when compared with *B. subtilis* and *L. fermentum*; the condition of *L. fermentum* also showed a significant difference compared with other strains. However, the study also showed that *B. subtilis* bacteria showed a high degree of sensitivity to *Gentamicin* (S ≥ 15 mm), and the remaining bacteria only showed an intermediate level of resistance (13 ≤ I ≤ 14 mm).

### 3.8. Evaluation of the Antibacterial Ability of Potential Probiotics Compared with Commercial Strains of Probiotics

The antibacterial ability of probiotic strains is one of the outstanding properties of probiotic bacteria for human health. The results of antibacterial activity based on the diameter of the inhibitor zone on agar plates are shown in Table 12. Experimental images are included in Appendix B. The results showed that all five probiotic strains had antibacterial activity against 10 strains of pathogenic bacteria, and the diameter was 9–14 mm. The *B. subtilis* strain showed the best antibacterial ability against two pathogenic bacteria (*C. jejuni* and *S. boydii*), with 15 mm ± 0.00 and 15 mm ± 1.00 inhibition zones, respectively. *L. plantarum*, *B. clausii*, and *L. fermentum* showed no differences in inhibitory ability in pathogenic strains. *L. acidophilus* showed the lowest inhibitory ability, among which *P. mirabilis* had the smallest inhibitory zone diameter of 7.00 mm ± 0.00. In general, the antibacterial activity of Gram-positive bacteria including *Bacillus cereus*, *Listeria monocytogenes*, and *Staphylococcus aureus* showed almost no differences compared to Gram-negative bacteria such as *E. coli*, *S. typhimurium*, *C. jejuni*, *V. parahaemolyticus*, *C. freundii*, *P. mirabilis*, and *S. boydii*. 

## 4. Discussion

This study showed that the N2 and N6 strains had similar results to the study by Ibrahim [25] on the isolation of LAB from mango and the study by Amelia [26] on isolates of beneficial LAB strains from Dadiah. Still, all showed that Gram-positive, rod-shaped LAB strains were able to use glucose and did not produce catalase enzymes. The identification results by the 16S rRNA sequencing method showed that the genes from the N2 strain were similar to the genes encoding 16S rRNA of the *Lactobacillus* fermentum strain (accession number MT611892.1), and N6 had 100% similarity with the *Lactiplantibacillus plantarum* strain (accession number OQ891485.1). Similar to the research results of Dung and Ly on isolating and selecting LAB capable of producing antibacterial substances, the study also showed that pickled juice contains many bacteria belonging to the genus *Lactobacillales* [27]. In addition, Sakai [28] also isolated *L. (para) plantarum* from takanazuke or fermented *Brassica juncea* (L.) Czern. et Coss.

The percentage of bacteria at low pH in this study is similar to that of Baloch [29] on the potential determination of *Lactiplantibacillus plantarum* Lp-1, which showed that all potential probiotics could survive at pH 3 for 4 h of incubation. In the study of Khalil [30] on the probiotic characterization of *Lactobacillus* strains, the authors concluded that seven *Lactobacillus* strains isolated together with two controlled strains showed the ability to survive under conditions of pH 3 in 3 h of incubation with strain *L. plantarum* (DUR8) achieving the highest survival rate of 90.24%. This can be explained by how at lower pH, the metabolism and the number of viable bacteria cells decrease more because HCl acid in the simulated stomach media is a potent oxidizing agent, which can oxidize critical chemical components in microbial cells [31].

Similarly, in the studies of Kaewnopparat [23] and Kang [24] on *L. fermentum* SK5 and *L. salivarius* probiotic strains in pepsin (3 g/L, at pH 3, pH 4), the results showed that after 3 h and 4 h of incubation, the survival rate of probiotics reached more than 99% or was barely affected. At pH 2, for the *L. fermentum* SK5 strain, the survival rate decreased after 3 h of incubation, and the number of viable cells reached 70.48%, while for the *L. salivarius* strain after 4 h, the survival rate was 85%. Thus, when the incubation time in the media with pepsin at pH 2 was 3 h or 4 h, the percentage of viable cells compared to the original decreased.

The percentage rate of vital bacteria in pancreatin media was similar to the results of Sidira [32]. *L. casein* probiotics survived in the media containing pancreatin (1 g/L, pH 8), showing that the cell survival rate was 86.24% after 4 h of incubation. In another study on the *L. salivarius* MG242 probiotic strain in pancreatin (1 g/L, pH 7) media after 4 h, the percentage rate of cell survival tended to decrease but was not affected much. According to the study of Masco [33], the presence of pancreatin in the small intestine does not seem to create an insurmountable barrier to the culture of probiotics, so in the pancreatin media, the percentage of the number of vital cells showed little or no decrease with incubation time.

The ability of the potential probiotics in bile salt media is like the study of Shokryazdan [34] when evaluating probiotic activity on 10 *Lactobacillus* strains in a simulated 0.3% bile salt media, of which 8 strains had a reduced survival rate and 2 strains had a slightly increased survival rate after 3 h of incubation. Furthermore, the study results of Zhang [35] evaluating the survival ability in 0.25% simulated bile salt media after 3 h of incubation on 19 *Lactobacillus* strains showed that all strains showed a decrease in the number of viable cells, and the survival rate was 50% higher. The authors Urdaneta and Casadesus [36] explain that bile salts also induce peptidoglycan remodeling on bacterial cell membranes, and remodeling increases bile resistance. In addition, deoxycholate (DOC) has below-lethal levels, reduces lipoprotein activity, and increases the cell membrane fluidity of microorganisms. The presence of DOC was also associated with a reduction in 3-3 crosslinking between the sugar components of peptidoglycan (N-acetylmuramic acid and N-acetylglucosamine), suggesting that low crosslinking may increase bile salt resistance.

The hydrophobicity of the bacteria of potential probiotics is consistent with the research of Tuo [37] investigating the aggregation and adhesion properties of 22 *Lactobacillus* strains, and *Lactiplantibacillus plantarum* 130 has a hydrophobic capacity of 26.71%. The study of Dhewa [38] on the hydrophobicity of *Lactobacillus* strains with three types of hydrocarbon (n-hexadecane, xylene, toluene) showed that the hydrophobicity of bacterial strains in each hydrophobicity type is different, in which the adhesion efficiency of probiotic strains in xylene solvent was in the range 12–73%. The intestinal wall is the place for probiotic bacteria, so the ability of probiotic strains to adhere to human intestinal cells is considered an essential prerequisite for the activity of probiotics. In conclusion, both strains of LAB isolated from pickled *Brassica juncea* (L.) Czern. et Coss. have hydrophobicity; however, *Lactiplantibacillus plantarum* has better hydrophobicity than *Lactobacillus fermentum*.

Several studies have also demonstrated that probiotic strains such as *Bacillus*, *Bifidobacterium*, and *Lactococcus* can all contain genotype-regulated antibiotic resistance genes, so different probiotics may exhibit specific characteristics [39]. This result is consistent with the study of Yến [40] when investigating the probiotic properties of *Bacillus subtilis* strains isolated in the Mekong Delta provinces; the results showed that 21 *Bacillus subtilis* strains were sensitive to most antibiotics, of which 100% of the strains of *B. subtilis* were susceptible to *Doxycycline* and *Enrofloxacin* and 24% of the *Bacillus* isolates were susceptible to the antibiotic *Gentamicin*.

The antibacterial results of the probiotic strains in the study also showed similar results to the study of Tharmaraj and Shah [41] on the antibacterial effect of probiotics against pathogenic bacteria (*Escherichia coli*, *Salmonella typhimurium*, *Staphylococcus aureus*, *Bacillus cereus*) and spoilage bacteria (*Bacillus stearothermophilus* and *Pseudomonas aeruginosa*) selected in cheese sauces. Similarly, the study by Coman [42] evaluating the resistance of pathogenic bacteria to LAB also showed that all were resistant to pathogenic bacteria at different concentrations. Research on the antibacterial activity of probiotic strains shows that, during growth, probiotic bacteria will produce organic compounds such as acetic acid, lactic acid, or propionic acid, which inhibit pathogenic bacteria. Resistance to pathogenic bacteria is based on the mechanism of entering bacterial cells through the cytoplasm, then breaking down into hydrogen ions, causing the acidification of the cytoplasm, thereby changing the metabolism, destroying enzymes, inhibiting protein biosynthesis, preventing the synthesis components of cell walls, disrupting nutrient absorption, and thereby destroying cell walls [43].

## 5. Conclusions

In this study, *Lactiplantibacillus plantarum* was isolated from pickled Vietnamese cabbage, with the percentage of viable cells incubated in pH 1 media for 90 min; pepsin 3 g/L media, pH 2 for 180 min; pancreatin media 1 g/L, pH 7 for 4 h; and bile salt 0.3% for 4 h. The hydrophobicity of the strain *Lactiplantibacillus plantarum* was better than some commercial strains; *Lactiplantibacillus plantarum* also showed sensitivity to the three antibiotics *Doxycycline*, *Enrofloxacin*, *Amoxicillin*, and intermediate sensitivity to *Gentamicin*; the inhibition zone diameter of *Lactiplantibacillus plantarum* was sensitive to some pathogenic bacteria. The five tested strains showed probiotic activity and the activity was ranked in descending order as follows: *Bacillus subtilis* > *Lactiplantibacillus plantarum* > *Lactobacillus acidophilus* > *Bacillus clausii* > *Lactobacillus fermentum*. From this, it can be seen that isolated probiotics from pickled cabbage can also be considered for use in food products.

## Figures and Tables

**Table 1 foods-12-03810-t001:** The number of bacterial strains on the surface of pickled *Brassica juncea* (L.) Czern. et Coss.

Source	Numbers of Bacterial Strains	Symbol
Go Vap Coopmart Supermarket	6	A1, A2, A3, A4, A5, A6
Go Vap Market	5	B1, B2, B3, B4, B5
Thu Duc Market	5	C1, C2, C3, C4, C5
Go Vap Emart Supermarket	6	D1, D2, D3, D4, D5, D6

**Table 2 foods-12-03810-t002:** Colony characteristics of potential probiotic strains.

Initial Colony Strains	Colony Strains with Similar Morphological Characteristics	Morphological Characteristics	Illustrating Images
A1, B2, C1, D4	N1	Colonies are pale yellow, relatively round, quite large, prominent on the surface of the agar plate, the surface is a membrane covering the inner liquid, the surface is not glossy, slightly wrinkled, and the cover is tooth-shaped light saw.	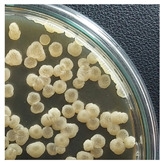
A5, B3, C2, B5	N2	Colonies are ivory-white, relatively round, small, floating on the surface of the agar plate, the surface and inside of the colonies are uniform, glossy surfaces, round covers.	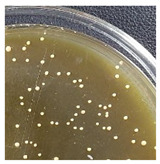
A2, B1, C4, D2	N3	Colonies are ivory white, rough surface, irregular round shapes, surface with raised small circles, uniform surface and inside, slightly serrated covers.	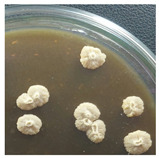
A3, C3, D3, B4	N4	Colonies are milky white, rough surface, slightly rounded shapes protruding above the surface of the plate, the surface and inside of the colony are uniform, round cover.	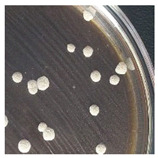
A4, C5, D5	N5	Colonies are ivory white, spherical, prominent on the surface of the agar plate, smooth surface, not glossy, uniform surface and inside, round cover.	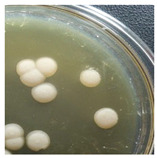
C6, D1, A6	N6	Colonies are ivory white, almost round, slightly raised on the agar surface, glossy and plump surface, and round cover.	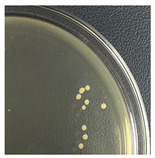

**Table 3 foods-12-03810-t003:** Morphological characteristics of bacterial strains on the surface of pickled *Brassica juncea* (L.) Czern. et Coss. when observed under a microscope (100×).

Colony Group	Morphological Characteristics	Image under a 100× Microscope
N1	Short rod-shaped, Gram-positive bacteria	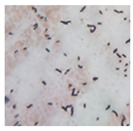
N2	Long rod-shaped, Gram-positive bacteria	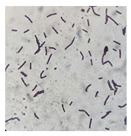
N3	Short rod-shaped, Gram-positive bacteria	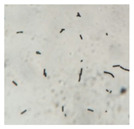
N4	Short rod-shaped, Gram-positive bacteria	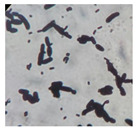
N5	Short rod-shaped, Gram-negative bacteria	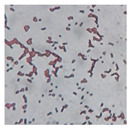
N6	Short rod-shaped, Gram-positive bacteria	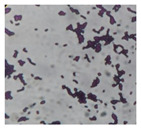

**Table 4 foods-12-03810-t004:** Results of the biochemical testing of potential bacterial strains.

Bacterial Strains	Catalase Test	Color Change of Glucose Solution (after 24 h Incubation with Bacteria and Bromocresol Green Reagent)	pH of Glucose Solution (after 24 h Incubation with Bacteria)
N1	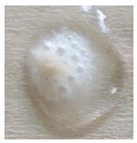 (+)	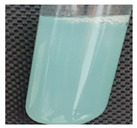	5.43
N2	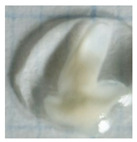 (-)	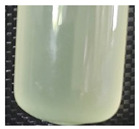	3.45
N3	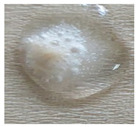 (-)	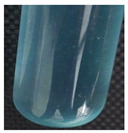	6.15
N4	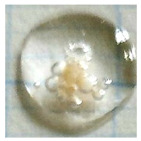 (+)	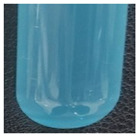	5.82
N5	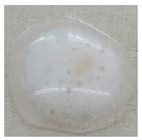 (+)	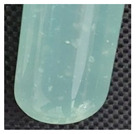	5.65
N6	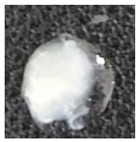 (+)	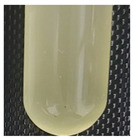	2.76

**Table 5 foods-12-03810-t005:** Identification results by 16S rRNA gene sequencing.

Accession	Description	Query Coverage (%)	E-Value	Max Identity (%)
MT611892.1	*Lactobacillus fermentum* strain 2953 16S ribosomal RNA gene, partial sequence	100	0.0	100
OQ891485.1	*Lactiplamtibacillus plantarum* strain KUMS-C47 16S ribosomal RNA gene, partial sequence	100	0.0	100

**Table 6 foods-12-03810-t006:** The percentage of bacterial cells that survived in the stomach media at pH 1, pH 2, and pH 3 at the survey time intervals of 30 min, 90 min, and 180 min (%).

	pH 1			pH 2			pH 3		
	30 min	90 min	180 min	30 min	90 min	180 min	30 min	90 min	180 min
*L. acidophilus*	51.57 ± 2.39 ^bD^	1.62 ± 0.01 ^cA^	0 ± 0 ^aA^	70.43 ± 3.61 ^bcE^	53.98 ± 0.86 ^cD^	24.51 ± 0.18 ^dB^	86.41 ± 0.12 ^dG^	77.43 ± 0.17 ^eF^	35.45 ± 0.06 ^dC^
*B. subtilis*	46.38 ± 1.52 ^bE^	4.04 ± 0.11 ^eB^	0 ± 0 ^aA^	76.37 ± 3.15 ^cH^	56.97 ± 0.38 ^dF^	28.35 ± 0.14 ^eC^	87.18 ± 0.10 ^eI^	65.29 ± 0.12 ^bG^	34.26 ± 0.23 ^cD^
*B. clausii*	30.46 ± 0.95 ^aC^	1.30 ± 0.07 ^bA^	0 ± 0 ^aA^	68.90 ± 1.60 ^bF^	50.93 ± 0.46 ^bE^	21.13 ± 0.07 ^bB^	80.18 ± 0.24 ^bG^	68.24 ± 0.19 ^dF^	37.54 ± 0.26 ^eD^
*L. fermentum*	28.25 ± 0.53 ^aE^	0.93 ± 0.01 ^aB^	0 ± 0 ^aA^	58.25 ± 0.22 ^aH^	43.9 ± 0.41 ^aF^	18.41 ± 0.09 ^aC^	67.17 ± 0.06 ^aI^	49.32 ± 0.08 ^aG^	27.41 ± 0.04 ^aD^
*L. plantarum*	34.52 ± 5.78 ^aC^	2.53 ± 0.14 ^dA^	0 ± 0 ^aA^	72.54 ± 0.45 ^bcE^	52.28 ± 1.18 ^bcD^	22.67 ± 0.40 ^cB^	83.11 ± 0.09 ^cF^	67.38 ± 0.06 ^cE^	33.42 ± 0.09 ^bC^

Lower-case letters a, b, c, etc., represent significant differences at a 5% level between bacterial strains when compared at the same pH value at different survey time intervals; capital letters A, B, C, etc., represent significant differences at a 5% level in the percentage of live bacteria of a bacterial strain at different pH values and survey times.

**Table 7 foods-12-03810-t007:** Percentage of bacterial cells that survived in pepsin media at the survey time intervals of 30, 90, and 180 min (%).

	Percentage Survival Rate of Bacteria in Pepsin Media (%)		
	30 min	90 min	180 min
*L. acidophilus*	61.33 ± 0.25 ^dC^	55.02 ± 0.00 ^eA^	47.45 ± 0.22 ^dB^
*B. subtilis*	52.12 ± 0.08 ^cC^	47.06 ± 0.04 ^cA^	43.19 ± 0.05 ^eB^
*B. clausii*	46.01 ± 0.01 ^bC^	37.08 ± 0.01 ^bA^	31.22 ± 0.04 ^bB^
*L. fermentum*	42.11 ± 0.09 ^aC^	25.04 ± 0.04 ^aA^	18.20 ± 0.06 ^aB^
*L. plantarum*	65.08 ± 0.10 ^eC^	49.04 ± 0.01 ^dA^	41.16 ± 0.14 ^cB^

The lower letters a, b, c, etc., represent significant differences at a 5% level in the percentage of viable cells between bacterial strains according to incubation time at 30 min, 90 min, and 180 min; the capital letters A, B, C, represent significant differences at a 5% level in the percentage of viable cells of a bacterial strain when incubated in pepsin media at different time intervals.

**Table 8 foods-12-03810-t008:** Percentage of bacterial cells that survived in pancreatin media at the survey time intervals of 1 h, 2 h, 3 h, and 4 h (%).

Percentage Survival Rate of Bacteria in Pancreatin Media (%)				
	1 h	2 h	3 h	4 h
*L. acidophilus*	96.35 ± 0.02 ^cC^	96.25 ± 0.00 ^cC^	95.27 ± 0.15 ^bB^	94.27 ± 0.07 ^bA^
*B. subtilis*	98.15 ± 0.04 ^eB^	98.57 ± 0.00 ^eC^	97.13 ± 0.11 ^dA^	97.13 ± 0.78 ^dA^
*B. clausii*	94.86 ± 0.01 ^aC^	94.18 ± 0.16 ^aB^	94.27 ± 0.04 ^aB^	93.15 ± 0.04 ^aA^
*L. fermentum*	95.19 ± 0.01 ^bB^	95.12 ± 0.08 ^bB^	95.2 ± 0.10 ^bB^	94.36 ± 0.05 ^bA^
*L. plantarum*	97.25 ± 0.02 ^dB^	97.24 ± 0.03 ^dB^	96.2 ± 0.12 ^cA^	96.2 ± 0.12 ^cA^

The lower letters a, b, c, etc., represent significant differences at a 5% level in the percentage of bacterial cells that survived among bacterial strains according to incubation times of 1 h, 2 h, 3 h, and 4 h; the capital letters A, B, C, represent significant differences at a 5% level in the percentage of surviving bacterial cells of a bacterial strain at different incubation times.

**Table 9 foods-12-03810-t009:** Percentage of bacterial cells that survived in bile salt media at the survey time intervals of 1 h, 2 h, 3 h, and 4 h (%).

Percentage Survival Rate of Bacteria in Bile Salt Media (%)				
	1 h	2 h	3 h	4 h
*L. acidophilus*	81.2 ± 0.04 ^cD^	73.42 ± 0.13 ^cC^	67.09 ± 0.08 ^cB^	56.2 ± 0.07 ^cA^
*B. subtilis*	92.19 ± 0.05 ^eD^	86.15 ± 0.03 ^eC^	75.19 ± 0.08 ^eB^	64.17 ± 0.05 ^eA^
*B. clausii*	78.12 ± 0.13 ^bD^	66.26 ± 0.09 ^bC^	57.24 ± 0.19 ^bB^	44.15 ± 0.07 ^bA^
*L. fermentum*	65.07 ± 0.04 ^aD^	52.13 ± 0.07 ^aC^	48.14 ± 0.08 ^dB^	39.27 ± 0.05 ^aA^
*L. plantarum*	89.42 ± 0.22 ^dD^	82.16 ± 0.07 ^dC^	76.11 ± 0.02 ^aB^	62.3 ± 0.06 ^dA^

The lower letters a, b, c, etc., represent significant differences at a 5% level in the percentage of bacterial cells that survived among bacterial strains according to incubation times of 1 h, 2 h, and 3 h, 4 h; the capital letters A, B, C, etc., represent significant differences at a 5% level in the percentage of surviving bacterial cells of a bacterial strain at different incubation times.

**Table 10 foods-12-03810-t010:** Hydrophobicity of 5 probiotic strains.

Bacteria	Percentage Average Hydrophobicity (%)
*B. subtilis*	40.42 ± 0.74 ^e^
*L. plantarum*	30.95 ± 0.91 ^d^
*B. clausii*	26.03 ± 0.31 ^c^
*L. acidophilus*	22.45 ± 1.27 ^b^
*L. fermentum*	18.67 ± 0.84 ^a^

Different letters represent significant differences at the 5% level in the percentage of hydrophobicity.

**Table 11 foods-12-03810-t011:** Inhibition ring diameter of 5 bacterial strains on 4 antibiotics *Doxycycline*, *Enrofloxacin*, *Gentamicin*, and *Amoxicillin*.

Bacteria	Inhibition Ring Diameter (mm)			
	Type of Antibiotic			
	*Doxycycline* 30 μg	*Enrofloxacin* 5 μg	*Gentamicin* 10 μg	*Amoxicillin* 10 μg
*B. subtilis*	29.67 ± 0.58 ^a^	25.00 ± 1.00 ^ab^	12.33 ± 0.58 ^a^	28.67 ± 0.58 ^b^
*L. plantarum*	30.33 ± 1.53 ^a^	23.00 ± 1.00 ^a^	13.33 ± 0.58 ^ab^	27.33 ± 1.59 ^b^
*B. clausii*	30.33 ± 1.53 ^a^	30.67 ± 1.53 ^c^	15.67 ± 1.53 ^b^	22.67 ± 1.16 ^a^
*L. acidophilus*	29.67 ± 1.53 ^a^	24.67 ± 1.52 ^ab^	14,67 ± 0.58 ^ab^	26.00 ± 1.00 ^ab^
*L. fermentum*	30.67 ± 1.55 ^a^	26.67 ± 1.15 ^b^	13.67 ± 1.16 ^ab^	27.00 ± 1.73 ^b^

Letters a, b, c represent significant differences at a 5% inhibition zone diameter when comparing 5 probiotics and 4 antibiotics.

**Table 12 foods-12-03810-t012:** Diameter of the inhibition zone of harmful bacteria by 5 probiotic strains.

Bacteria	Inhibition Zone Diameter (mm)					
	*L. acidophilus*	*B. clausii*	*B. subtilis*	*L. fermentum*	*L. plantarum*	*Ciprofloxacin*
*B. cereus*	11.00 ± 1.00 ^a^	12.33 ± 0.57 ^ab^	14.66 ± 0.57 ^b^	12.66 ± 0.57 ^ab^	12.33 ± 0.57 ^ab^	24.00 ± 1.73 ^c^
*C. jejuni*	9.66 ± 1.15 ^a^	13.33 ± 1.15 ^bc^	15.00 ± 0.00 ^c^	12.00 ± 1.00 ^ab^	14.00 ± 0.00 ^bc^	27.33 ± 1.52 ^c^
*C. freundii*	9.33 ± 1.15 ^a^	12.00 ± 1.00 ^ab^	13.33 ± 0.57 ^b^	11.66 ± 0.57 ^ab^	12.00 ± 1.00 ^ab^	23.00 ± 1.73 ^c^
*E. coli*	10.66 ± 0.57 ^a^	11.66 ± 0.57 ^a^	15.66 ± 0.57 ^b^	12.00 ± 1.00 ^ab^	11.33 ± 1.52 ^a^	30.00 ± 0.00 ^c^
*L. monocytogenes*	9.00 ± 1.00 ^a^	11.66 ± 0.57 ^b^	14.66 ± 0.57 ^c^	12.00 ± 0.00 ^b^	12.33 ± 0.57 ^b^	23.00 ± 1.73 ^c^
*P. mirabilis*	7.00 ± 0.00 ^a^	9.00 ± 1.00 ^b^	13.66 ± 1.15 ^d^	11.33 ± 0.57 ^c^	11.66 ± 0.57 ^c^	28.33 ± 1.52 ^e^
*S. boydii*	7.66 ± 0.57 ^a^	11.66 ± 0.57 ^b^	15.00 ± 1.00 ^c^	11.00 ± 1.00 ^b^	12.66 ± 1.52 ^bc^	26.00 ± 1.73 ^d^
*S. typhimurium*	10.00 ± 0.00 ^a^	11.00 ± 1.00 ^ab^	13.66 ± 0.57 ^c^	12.00 ± 1.00 ^b^	12.33 ± 0.57 ^bc^	30.00 ± 0.00 ^d^
*S. aureus*	9.33 ± 0.57 ^a^	12.00 ± 0.00 ^a^	13.66 ± 0.57 ^a^	12.66 ± 1.15 ^a^	12.33 ± 1.52 ^a^	23.00 ± 3.46 ^b^
*V. parahaemolyticus*	9.33 ± 0.57 ^a^	12.00 ± 1.00 ^b^	14.33 ± 0.57 ^c^	13.33 ± 0.57 ^bc^	12.00 ± 1.00 ^b^	21.00 ± 0.00 ^d^

Pathogen abbreviation: *B. cereus* (*Bacillus cereus* NRRL B-3711); *C. jejuni* (*Camplyobacter jejuni* DSM 24114); *C. freundii* (*Citrobacter freundii* RLS1); *E. coli* (*Escherichia coli* NRRL B-409); *L. monocytogenes* (*Listeria monocytogenes* CIP 106); *P. mirabilis* (*Proteus mirabilis* ADL-72); *S. boydii* (*Shigella boydii* ATCC8700); *S. typhimurium* (*Salmonella typhimurium* DSM 10506); *S. aureus* (*Staphylococcus aureus* NRRL B-313); *V. parahaemolyticus* (*Vibrio parahaemolyticus* DSM2172). Letters a, b, c, etc., represent significant differences at a 5% inhibition zone diameter when comparing the antibacterial ability of 5 probiotic strains.

## Data Availability

Data is contained within the article.

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
