# Peer review of "Isolation and Evaluation of the Probiotic Activity of Lactic Acid Bacteria Isolated from Pickled Brassica juncea (L.) Czern. et Coss."

_foods, 2023, doi:10.3390/foods12203810_

Round 1
Reviewer 1 Report
The manuscript entitled “Isolation and evaluation of probiotic activity of lactic acid bacteria isolated from pickled Brassica Juncea (L.) Czern. et Coss.” refers to very important and interesting issue.
Article deals with an isolation lactic acid bacteria form pickled Brassica Juncea (L.) Czern. et Cos.
The two isolated strains, such as Lactobacillus fermentum and Lactiplantibacillus plantarum isolated from pickled Vietnamese cabbage showed probiotic properties similarly as commercial probiotic strains of Bacillus subtilis, Bacillus clausii, Lactobacillus acidophilus. Authors evaluated the ability to survive in low pH, pepsin, pancreatin, and bile salt media, hydrocarbon adhesion ability, antibiotic resistance, and resistance to pathogenic bacteria of all tested strains. They observed that the isolated strain Lactiplantibacillus plantarum has similar properties as commercial strains.
The provided results fully illustrate the problem under consideration.
I think that the manuscript should be published after minor correction and explanation.
I have some doubt refers to the applied statistical analysis.
1. Was the normality of the distribution of the variable checked, the Shapiro-Wilk test? I would suggest using the analysis of variance with the software STATISTICA To compare the five data sets. This will allow a simultaneous reliable comparison.
2. Why did the percentage of bacterial cells that survived in the stomach media at pH 1, pH 2, and pH 3 measure in time intervals 30 min, 90 min, and 180 min? It is according to some standard?
Author Response
Response to Reviewer 1 Comments |
||
1. Summary |
|
|
We would like to thank you for the suggested revisions to clarify the issue in our research. |
||
2. Questions for General Evaluation |
Reviewer’s Evaluation |
Response and Revisions |
Does the introduction provide sufficient background and include all relevant references? |
Can be improved |
|
Are all the cited references relevant to the research? |
Yes |
|
Is the research design appropriate? |
Can be improved |
|
Are the methods adequately described? |
Can be improved |
|
Are the results clearly presented? |
Can be improved |
|
Are the conclusions supported by the results? |
Yes |
|
3. Point-by-point response to Comments and Suggestions for Authors |
||
Comments 1: Was the normality of the distribution of the variable checked, the Shapiro-Wilk test? I would suggest using the analysis of variance with the software STATISTICA To compare the five data sets. This will allow a simultaneous reliable comparison? |
||
Response 1: Thank you for pointing this out. In this article, I used ANOVA statistical analysis with Tukey's Test using SPSS 20 software to evaluate the significant or non-significant differences of the results obtained in 5 data sets. I have understood that the Shapiro-Wilk test determines whether a sample is likely to come from a normal distribution by testing its normality. In order to perform several of the well-known statistical tests, like the Student's t-test and ANOVA, the normality of the distribution must be confirmed. I would like to thank you for your guidance on STATISTICA software, which I will use in my upcoming research. |
||
Comments 2: Why did the percentage of bacterial cells that survived in the stomach media at pH 1, pH 2, and pH 3 measure in time intervals 30 min, 90 min, and 180 min? It is according to some standard? |
||
Response 2: Thank you for pointing this out. Generally, food stays in the stomach for 40 minutes to 2 hours before moving on to the small intestine. After that, it stays in the small intestine for around five hours before passing through the colon. Therefore, the study evaluated the % of bacterial cells surviving in simulated gastric media in the initial period of 30 minutes, then the final period of 90 minutes and 180 minutes without evaluating in 60 minutes. Besides, Tulini et al., 2013 survial of L. paraplantarum FT259 exposed to low pH in 0 minute and 180 minutes. Latha et al., 2016 also used 0 and 180 minutes to survey actinobacteria in low pH. 4. Response to Comments on the Quality of English Language Point 1: English language fine. No issues detected Response 1: Thank you for your comment. The manuscript has been corrected for some minor errors. 5. Additional clarifications No addition |
Reviewer 2 Report
The article is devoted to the characterization of lactic acid bacteria extracted from pickled Vietnamese cabbage. The researchers' goal was to prove that these extracted bacteria could be used as a good prebiotic. The article contains a lot of microbiological data obtained using classical methods. The authors included microphotographs of bacteria, which of course increased the attractiveness of the article.
The article can be improved considering the following comments and suggestions:
- The authors must explain in the background why, among the 3500 foods containing lactic acid bacteria (authors give this number in line 33), the authors chose namely pickled Vietnamese cabbage.
- When the authors give data on the hydrophobicity of bacteria, they give several names – H (line 142), hydrophobicity (line 132), affinity for hydrocarbons (line 141), hydrophobic capacity (255), hydrophobic performance (266), although they are talking about the same thing. Authors have to choose one, otherwise readers may get confused.
- If the authors want to express the values of H as a percentage, then they need to multiply the right side of equation (1) by 100%. Therefore 100% must be inserted into the equation.
- The accuracy of the value determining in this type of research is 3 or 4 significant digits. Therefore, there is no point in giving more digits. For example, in Table 10 the values 40.4±0.7 should be given instead of 40.424±0.74 and 31.0±0.9 instead of 30.951±0.91.
-Authors must correct typos in the text. For example, in several places links are given without square brackets.
Author Response
Response to Reviewer 2 Comments
|
||
1. Summary |
|
|
We would like to thank you for the suggested revisions to clarify the issue in our research. |
||
2. Questions for General Evaluation |
Reviewer’s Evaluation |
Response and Revisions |
Does the introduction provide sufficient background and include all relevant references? |
Can be improved |
|
Are all the cited references relevant to the research? |
Yes |
|
Is the research design appropriate? |
Can be improved |
|
Are the methods adequately described? |
Can be improved |
|
Are the results clearly presented? |
Can be improved |
|
Are the conclusions supported by the results? |
Yes |
|
3. Point-by-point response to Comments and Suggestions for Authors |
||
Comments 1: The authors must explain in the background why, among the 3500 foods containing lactic acid bacteria (authors give this number in line 33), the authors chose namely pickled Vietnamese cabbage. |
||
Response 1: Pickled Brassica juncea (L.) Czern. et Coss. was used in this study because this is a very popular food ingredient in Vietnam that contains a high source of lactic bacteria, is easy to ferment and is often used to eat raw or processed. |
||
Comments 2: When the authors give data on the hydrophobicity of bacteria, they give several names – H (line 142), hydrophobicity (line 132), affinity for hydrocarbons (line 141), hydrophobic capacity (255), hydrophobic performance (266), although they are talking about the same thing. Authors have to choose one, otherwise readers may get confused |
||
Response 2: The term used for consistency in referring to the hydrophobic properties of bacteria showed in the manuscript was chosen as hydrophobicity. In the manuscript, lines 67, 142, 144, 255, 262, 269, 373, 376 were edited and highlighted. Comments 3: If the authors want to express the values of H as a percentage, then they need to multiply the right side of equation (1) by 100%. Therefore 100% must be inserted into the equation. Response 3: The value was expressed as % and 100 has been added to the equation on line 145. Comments 4: The accuracy of the value determining in this type of research is 3 or 4 significant digits. Therefore, there is no point in giving more digits. For example, in Table 10 the values 40.4±0.7 should be given instead of 40.424±0.74 and 31.0±0.9 instead of 30.951±0.91. Response 4: The significant decimal digits adjusted throughout the article were 2 digits. Comments 5: Authors must correct typos in the text. For example, in several places links are given without square brackets. Response 5: Spelling errors and reference citation errors have been corrected in the manuscript. 4. Response to Comments on the Quality of English Language Point 1: I am not qualified to assess the quality of English in this paper Response 1: Thank you for your comment. The original copy has been redressed for a few minor blunders. 5. Additional clarifications No addition |
Reviewer 3 Report
Rewiew Manuscript ID: foods-2626686
Brief summary: In the paper entitled "ISOLATION AND EVALUATION OF PROBIOTIC ACTIVITY 2 OF LACTIC ACID BACTERIA ISOLATED FROM THE DECAPATE 3 OF BRASSICA JUNCEA (L.) CZERN. ET COSS.", the Authors compared the probiotic properties of probiotics isolated from Vietnamese pickled cabbage with some commercial probiotic strains found in the Vietnamese market. The results showed that 2 strains (Lactobacillus fermentum and Lactiplantibacillus plantarum) isolated from Vietnamese pickled cabbage and 3 commercial probiotic strains (Bacillus subtilis, Bacillus clausii, Lactobacillus acidophilus) showed probiotic properties.
Probiotic properties were assessed by the ability to survive in low pH environments, pepsin, pancreatin and bile salts, hydrocarbon adhesion, antibiotic resistance and resistance to pathogenic bacteria. In particular, the isolated strain Lactiplantibacillus plantarum had probiotic properties lower than Bacillus subtilis but better than the two commercial strains Bacillus clausii, Lactobacillus acidophilus and isolated Lactobacillus fermentum showed the lowest probiotic properties of the five strains.
The paper meets the aims of the journal. However, it is not suitable for its current form.
Introduction: In accordance with journal guidelines, including appropriate bibliographic references, it would be appropriate to better outline the aim of the study.
Materials and methods: this section needs to be improved, both in the description of the methods and in the statistical analysis.
Results and the discussion need major improvements, including the tables.
The conclusions need to be rewritten and it is not advisable to report numerical values (e.g. mean values), which should be hidden in the results and should be commented on in the discussion.
References are incomplete.
Here are my main observations, line by line:
Line 50: References should be enclosed in square brackets, such as [10]. Check the text.
Line 52: CFU? Enter acronym (Colony Forming Unit).
Lines 60-64: rewrite in a clear way.
Line 65-152: “2. Materials and methods” need to be clearly rewritten. The formula in line 142 needs to be better explained.
Line 163-166: “2.2.9. Statistical analysis", describe the method used and the expression of the results.
Result and discussion: Write clearly, in tables use superscript to indicate significance. Table 8, % survival rate of bacteria in pancreatin media, use the symbol '%' at the end of the sentence, also check in the tables.
Conclusion should be rewritten by limiting numerical references.
References: report references should follow MDPI style.
There is an extensive English language editing requirement.
Author Response
Please see the attached file
Response to Reviewer 3 Comments |
||
1. Summary |
|
|
we would like to thank you for the suggested revisions to clarify the issue in our research. |
||
2. Questions for General Evaluation |
Reviewer’s Evaluation |
Response and Revisions |
Does the introduction provide sufficient background and include all relevant references? |
Can be improved |
|
Are all the cited references relevant to the research? |
Can be improved |
|
Is the research design appropriate? |
Can be improved |
|
Are the methods adequately described? |
Must be improved |
|
Are the results clearly presented? |
Must be improved |
|
Are the conclusions supported by the results? |
Must be improved |
|
3. Point-by-point response to Comments and Suggestions for Authors |
||
Comments 1: Line 50: References should be enclosed in square brackets, such as [10]. Check the text. |
||
Response 1: Spelling errors and reference citation errors have been corrected in the manuscript. |
||
Comments 2: Line 52: CFU? Enter acronym (Colony Forming Unit) |
||
Response 2: Spelling errors have been corrected (Colony Forming Unit per milliliter) Comments 3: Line 65-152: “2. Materials and methods” need to be clearly rewritten. The formula in line 142 needs to be better explained. Response 3: Additional explanations have been added to manucript and high light Comments 4: Line 163-166: “2.2.9. Statistical analysis", describe the method used and the expression of the results Response 4: The statistical analysis method has been outlined in more detail in the manuscript. Comments 5: Result and discussion: Write clearly, in tables use superscript to indicate significance. Table 8, % survival rate of bacteria in pancreatin media, use the symbol '%' at the end of the sentence, also check in the tables. Response 5: The symbol '%' has been adjusted in manuscript. Comments 6: Conclusion should be rewritten by limiting numerical references Response 6: Conclusions have been edited and data references limited. |
||
4. Response to Comments on the Quality of English Language |
||
Point 1: Extensive editing of English language required |
||
Response 1: The Quality of English Language has been edited and checked in the manucript again. |
||
5. Additional clarifications |
||
No addition |
Round 2
Reviewer 3 Report
Dear Authors, the following are my comments:
Lines 27-67: Standardise font throughout text.
Lines 178-182: I suggest the authors rewrite the sentences as follows: The results of the comparison and selection experiments were statistically processed and analysed using the IBM SPSS Statistics 20 program. Analysis of variance (ANOVA) and post hoc Tukey's test were used for group comparisons. The level of statistical significance was set at P < 0.05. Data are presented as mean ± standard deviation (SD).
Tables 6, 7, 8, 9, 10, 11, 12: The letters indicating the significant differences are in the following format, for example; 51.57±2.39 bD
Lines 420-432: Conclusions are to be formulated with the omission of the numerical content, which is already presented in the results section and in the tables
References: must be reported in MDPI style.
example reference n 20 (MDPI style)
Liu, S.; Zhang, Q.; Li, H.; Qiu, Z.; Yu, Y. Comparative Assessment of the Antibacterial Efficacies and Mechanisms of Different Tea Extracts. Foods 2022, 11, 620. https://doi.org/10.3390/foods11040620
All references (from n 1 to n. 43) must be adapted to MDPI style.
- I believe that a thorough revision of the English language is appropriate.
Author Response
Response to Reviewer 3 Comments |
||
1. Summary |
|
|
we would like to thank you for the suggested revisions to clarify the issue in our research. |
||
2. Questions for General Evaluation |
Reviewer’s Evaluation |
Response and Revisions |
Does the introduction provide sufficient background and include all relevant references? |
Can be improved |
|
Are all the cited references relevant to the research? |
Can be improved |
|
Is the research design appropriate? |
Can be improved |
|
Are the methods adequately described? |
Can be improved |
|
Are the results clearly presented? |
Can be improved |
|
Are the conclusions supported by the results? |
Can be improved |
|
3. Point-by-point response to Comments and Suggestions for Authors |
||
Comments 1: Lines 27-67: Standardise font throughout text. |
||
Response 1: Lines 27 – 66 were edited format for suitable with the format mdpi style |
||
Comments 2: Lines 178-182: I suggest the authors rewrite the sentences as follows: The results of the comparison and selection experiments were statistically processed and analysed using the IBM SPSS Statistics 20 program. Analysis of variance (ANOVA) and post hoc Tukey's test were used for group comparisons. The level of statistical significance was set at P < 0.05. Data are presented as mean ± standard deviation (SD). |
||
Response 2: We have revised the statistical analysis content according to the reviewer's suggestion. |
||
Comments 3: Tables 6, 7, 8, 9, 10, 11, 12: The letters indicating the significant differences are in the following format, for example; 51.57±2.39 bD Response 3: Letters represent significant differences of values in Tables 6, 7, 8, 9, 10, 11, 12 were edited. Comments 4: Lines 420-432: Conclusions are to be formulated with the omission of the numerical content, which is already presented in the results section and in the tables Response 4: The conclusion has been rewritten to show the research content and not mention the data. Comments 5: References: must be reported in MDPI style Response 5: Reference list were edited to follow MDPI style. |
||
4. Response to Comments on the Quality of English Language |
||
Point 1: Extensive editing of English language required |
||
Response 1: The Quality of English Language has been edited and checked in the manucript again. |
||
5. Additional clarifications |
||
No addition |